# The *BCO2* Genotype and the Expression of *BCO1*, *BCO2*, *LRAT,* and *TTPA* Genes in the Adipose Tissue and Brain of Rabbits Fed a Diet with Marigold Flower Extract

**DOI:** 10.3390/ijms24032304

**Published:** 2023-01-24

**Authors:** Janusz Strychalski, Andrzej Gugołek, Edyta Kaczorek-Łukowska, Zofia Antoszkiewicz, Paulius Matusevičius

**Affiliations:** 1Department of Fur-Bearing Animal Breeding and Game Management, University of Warmia and Mazury in Olsztyn, Oczapowskiego 5, 10-719 Olsztyn, Poland; 2Department of Microbiology and Clinical Immunology, Faculty of Veterinary Medicine, University of Warmia and Mazury in Olsztyn, Oczapowskiego 13, 10-719 Olsztyn, Poland; 3Department of Animal Nutrition and Feed Science, University of Warmia and Mazury in Olsztyn, Oczapowskiego 5, 10-719 Olsztyn, Poland; 4Department of Animal Breeding and Nutrition, Faculty of Animal Husbandry Technology, Lithuanian University of Health Sciences, Tilžes 18, LT-47181 Kaunas, Lithuania

**Keywords:** *BCO2* gene, rabbits, marigold, gene expression, carotenoids

## Abstract

This study was undertaken to evaluate the effect of the *BCO2* genotype and dietary supplementation with marigold flower extract on the expression of *BCO1*, *BCO2*, *LRAT,* and *TTPA* genes in the adipose tissue and brain of rabbits. The concentrations of lutein, zeaxanthin, β-carotene, retinol, and α-tocopherol were determined in samples collected from rabbits. Sixty young male Termond White rabbits were allocated to three groups based on their genotype at codon 248 of the *BCO2* gene (ins/ins, ins/del, and del/del). Each group comprised two subgroups; one subgroup was administered a standard diet, whereas the diet offered to the other subgroup was supplemented with 6 g/kg of marigold flower extract. The study demonstrated that the *BCO2* genotype may influence the expression levels of the *BCO2*, *LRAT,* and *TTPA* genes in adipose tissue, and *TTPA* and *BCO1* genes in the brain. Moreover, an increase in the amount of lutein in the diet of BCO2 del/del rabbits may increase the expression of *BCO1*, *LRAT,* and *TTPA* genes in adipose tissue, and the expression of the *BCO2* gene in the brain. Another finding of the study is that the content of carotenoids and α-tocopherol increases in both the adipose tissue and brain of *BCO2* del/del rabbits.

## 1. Introduction

Carotenoids, which belong to the isoprenoids, are synthesized by photosynthetic as well as non-photosynthetic organisms such as algae, bacteria, and fungi [1]. More than 750 structurally different carotenoids have been isolated to date. There are two main classes of carotenoids: Carotenes (e.g., α-carotene, β-carotene, and lycopene) and xanthophylls (e.g., lutein, zeaxanthin, and β-cryptoxanthin) [2]. During photosynthesis, carotenoids act as accessory pigments. They protect plants against photooxidative damage and serve as precursors for the biosynthesis of phytohormones [3,4]. Carotenoids give yellow, orange, and red colors not only to plants, fruits, and vegetables but also to insects, birds, and fish [5]. In animals, carotenoids are precursors to vitamin A, which plays a key role in eye health and vision, cellular growth, and differentiation [6]. In both plants and animals, carotenoids are potent antioxidants, effective singlet oxygen quenchers, and scavengers of other reactive oxygen species (ROS). They contribute to preventing many ROS-mediated disorders [7]. Animals are not able to synthesize carotenoids de novo, which is why they must be obtained from food [1]. Dietary carotenoids are absorbed and stored mainly in the liver and adipose tissue. They are metabolized with the involvement of digestive enzymes (pancreatic lipase and carboxylesterase) and enzymes that participate in their distribution to the target tissues (glutathione S-transferase Pi1, a protein homologous to StAR) [8,9,10]. Carotenoids are cleaved by two homologous enzymes: β-carotene oxygenase 1 (BCO1) and β-carotene oxygenase 2 (BCO2). Provitamin A carotenoids with unsubstituted β rings (such as α-carotene, β-carotene, and β-cryptoxanthin) can be cleaved by both enzymes. However, BCO2 has a higher affinity for carotenoids that do not serve as substrates for the biosynthesis of provitamin A (they do not have a β ring or have substituents at the β ring position, usually carbonyl, hydroxyl, or epoxy groups; examples of such carotenoids are lycopene, lutein, and zeaxanthin) [11]. BCO2 catalyzes the oxidative cleavage of carotenoids at the 9,10′ double bond, generating apo-10′-carotenals and ionones, whose functions remain largely unknown [12]. In turn, BCO1 converts β-carotene to retinal by centric oxidative cleavage at the 15,15′ double bond [13]. Retinal is then converted to retinol in a reaction catalyzed by retinol dehydrogenase. Next, lecithin retinol acyltransferase (LRAT) catalyzes the esterification of retinol to retinyl esters [14]. 

At slaughter, the fat of cattle, sheep, and rabbits can be assessed as white by a visual inspection. However, members of the above species may have a dysfunction in the BCO2 protein. As a result, non-provitamin A carotenoids are not cleaved; instead, they are deposited in the adipose tissue and impart a yellow color to fat [15,16,17]. Yellow fat is also often encountered in poultry [18]. Rabbits with the deletion of AAT nucleotides at codon 248 of the *BCO2* gene have yellowish fat [19]. The yellow tint of fat is a recessive trait typical of homozygous individuals—carriers of the above deletion [16,19,20].

Marigold flowers are abundant in carotenoid pigments. The Aztec marigold (*Tagetes erecta* L.) belongs to the family *Asteraceae.* These annual plant species are native to Mexico, but it is also widely grown in Europe, Asia, and Africa. Marigolds are ornamental flowers, but they also have numerous medicinal and industrial uses [21]. Lutein is the major pigment in the petals of marigold flowers, accounting for almost 90% of total carotenoids. In turn, zeaxanthin makes up around 5% of the carotenoids present in marigold flowers [22,23]. Poultry diets are often supplemented with marigold powder to enhance the color of the birds’ fat, skin, and egg yolk [21]. Dried alfalfa is often the main source of carotenoids in commercial pelleted diets for rabbits [20,24]. However, fresh alfalfa and other forages, which are often fed in large quantities to rabbits raised on small backyard farms, have considerably higher carotenoid content [25].

Our previous studies have investigated the levels of carotenoids, retinol, and α-tocopherol in the liver and adipose tissue of rabbits and the milk of does, as dependent on their genotype determined by AAT-deletion mutation at codon 248 of the *BCO2* gene [16,19,20]. A recent study evaluated the influence of the *BCO2* genotype and Aztec marigold flower extract added to rabbit diets on the expression of genes associated with the metabolism of carotenoids, vitamin A, and vitamin E (*BCO1*, *BCO2*, *LRAT*, and *TTPA* - α-tocopherol transfer protein) in the liver [26]. However, the responses of other tissues and organs have not been analyzed to date. Thus, to contribute to a more comprehensive understanding of carotenoid metabolism in rabbits, the present study was undertaken to analyze the effect of the *BCO2* genotype and dietary supplementation with marigold extract on the expression of *BCO1*, *BCO2*, *LRAT*, and *TTPA* genes in the adipose tissue and brain of rabbits. The levels of lutein, zeaxanthin, β-carotene, retinol, and α-tocopherol were also determined in the collected samples.

## 2. Results

Genotype at codon 248 of the *BCO2* gene differentiated the expression levels of *LRAT* and *TTPA* genes in the adipose tissue of rabbits receiving the control diet (Figure 1). The relative expression level of the *LRAT* gene was higher in ins/del than in del/del animals (*p* < 0.05), and the relative expression level of the *TTPA* gene was higher in ins/ins than in del/del animals (*p* < 0.05). In turn, in rabbits administered a diet containing 6 g/kg of marigold flower extract, the *BCO2* genotype influenced the differentiation of *BCO2* gene expression (Figure 2). Animals with the *BCO2* del/del genotype were characterized by a lower expression of this gene than ins/ins and ins/del individuals. However, no differences were found between the ins/ins and ins/del genotypes.

Dietary supplementation with marigold flower extract increased the expression of the *TTPA* gene in the adipose tissue of rabbits with *BCO2* ins/del and del/del genotypes (*p* < 0.01 in both cases) (Figure 3 and Figure 4). Moreover, dietary marigold supplementation increased the expression of *BCO1* and *LRAT* genes in rabbits carrying the *BCO2* del/del genotype (*p* < 0.05 in both cases) (Figure 4).

In perirenal fat, the levels of all tested micronutrients excluding retinol were affected by the *BCO2* genotype (Table 1). The content of xanthophylls (lutein and zeaxanthin) was many-fold higher in rabbits with the del/del genotype than in the animals with the remaining two genotypes (*p* <0.01 in all cases), and the noted differences were much greater in animals fed the marigold-supplemented diet (more than hundred-fold for lutein and more than fifty-fold for zeaxanthin) than in non-supplemented individuals (nine-fold and five-fold, respectively). Similarly, higher levels of β-carotene and α-tocopherol were observed in rabbits with the del/del genotype (vs. ins/ins and ins/del genotypes), and the noted differences were more pronounced in animals fed the fortified diet. Importantly, feeding marigold extract to rabbits with the del/del genotype induced a nine-fold increase in lutein concentration (*p* < 0.01), an eleven-fold increase in zeaxanthin concentration (*p* < 0.01), and a two-fold increase in β-carotene concentration (*p* < 0.05).

In rabbits fed the control diet, the relative expression level of the *TTPA* gene in the brain was higher in *BCO2* ins/del than in del/del (*p* < 0.01) animals (Figure 5), whereas in rabbits fed the marigold-enriched diet, a higher expression level of the *BCO1* gene was noted in heterozygous *BCO2* ins/del individuals compared to their counterparts with the ins/ins genotype (*p* < 0.05) (Figure 6).

As shown in Figure 7, heterozygous *BCO2* ins/del rabbits fed the fortified diet exhibited a higher expression of the *LRAT* gene in the brain than control group animals (*p* < 0.01). In turn, a higher expression of the *BCO2* gene (*p* < 0.01) was observed in the brain of homozygous del/del rabbits from the marigold group, compared with control group animals (Figure 8).

*BCO2* genotype and dietary xanthophyll concentrations affected the differentiation of carotenoid and α-tocopherol levels in the rabbits’ brains (Table 2). Animals with the *BCO2* del/del genotype were characterized by higher concentrations of these micronutrients than rabbits with ins/ins and ins/del genotypes. Overall, the intergroup relationships were similar to those found in adipose tissue, but the noted differences were smaller. Marigold extract added to rabbit diets increased the differences in the levels of lutein, zeaxanthin, and β-carotene in the brain between rabbits with the *BCO2* del/del genotypes and carriers of the other two genotypes. However, a lower concentration of α-tocopherol was noted in del/del rabbits fed the marigold-supplemented diet than in those fed the control diet (*p* < 0.01).

## 3. Discussion

To the best of the authors’ knowledge, this is the first study to evaluate changes in the expression levels of genes associated with the metabolism of carotenoids, vitamin A, and vitamin E in the adipose tissue and brain of rabbits. Moreover, we determined the rates of carotenoid, retinol, and α-tocopherol accumulation in the adipose tissue and, for the first time, in the brain of rabbits. The animals were allocated to three groups based on their *BCO2* genotype (ins/ins, ins/del, and del/del), and then each genotype was further divided into two subgroups based on dietary lutein level (the control diet contained 21.64 mg/kg of lutein, the marigold diet contained almost 320 mg/kg of lutein).

A comparison of the expression levels of the analyzed genes in the liver [26] vs. the adipose tissue and brain of rabbits (present study) revealed certain differences. The expression level of the *BCO1* gene was higher in the liver of heterozygous *BCO2* ins/del rabbits fed a typical farm-made (control) diet than in the liver of ins/ins rabbits [26], whereas the differences in the expression of this gene in adipose tissue and the brain were not significant (Figure 1 and Figure 5). However, the expression of the *LRAT* and *TTPA* genes in adipose tissue varied depending on both the rabbits’ *BCO2* genotype and dietary xanthophyll levels. The *LRAT* gene encodes an enzyme that transfers the acyl group from the sn-1 position of phosphatidylcholine to all-trans retinol, generating all-trans retinyl esters (the storage form of vitamin A) [27]. The *TTPA* gene encodes a transfer protein that binds α-tocopherol, with high selectivity and affinity, and transfers it between separate membranes [28]. The above genes are expressed not only in the liver but also in the brain, spleen, lung, kidneys, intestines, uterus and placenta, and peripheral tissues [29,30]. Thus, it appears that the levels of vitamins A and E in the adipose tissue and brain of rabbits could be modulated not only by the liver but also by local processes. This is natural since tissues and organs perform different functions. The liver is responsible for the synthesis of biochemicals and proteins; this excretory organ consists mostly (in 60–70%) of hepatocytes [31]. Cells of the brain, neurons, are electrically excitable and able to process and transmit information in the form of electrical signals [32]. Adipose tissue is composed of fat cells known as adipocytes. Its main role in mammals is to store energy and provide insulation. However, it should be noted that adipose tissue also participates in metabolism by secreting bioactive molecules (adipokines) including leptin, adiponectin, resistin, TNF, IL-6, IL-10, chemokine ligand 2, and TGFβ [33].

Our previous study [26] demonstrated that an increase in dietary lutein content could increase the expression of the *BCO2* gene in the liver of rabbits with the *BCO2* del/del genotype. Similar observations were made when the *BCO2* gene expression in the brain was analyzed (Figure 8). It should be noted that in *BCO2* del/del animals, dietary supplementation with marigold extract did not affect the expression level of this gene in adipose tissue, whereas the expression of *BCO1*, *LRAT,* and *TTPA*, increased (Figure 4). The antioxidant properties of carotenoids and vitamins A and E have been extensively researched [34], and it appears that their excess may be harmful [35]. It cannot be ruled out that the increased expression of the above genes in *BCO2* del/del animals constituted a line of defense against the excessive dietary intake of antioxidant compounds. Possibly, the above genes could produce different expression patterns if the examined animals were subjected to pro-oxidative treatment. This issue may be addressed in our future research.

Previous research has shown that the concentrations of lutein and β-carotene in adipose tissue are higher in *BCO2* del/del rabbits than in their counterparts with other genotypes (ins/ins and ins/del) [16,20,36]. Upon absorption, a major part of β-carotene is cleaved by BCO1, and the remaining part is either cleaved by BCO2 or accumulated in adipose tissue [13]. Therefore, functional impairment of BCO2 should be associated with an increase in both xanthophyll and β-carotene levels. Such a relationship was also observed in the present study, in adipose tissue and the brain. β-carotene is a precursor of vitamin A; therefore, its increased accumulation in the body may enhance vitamin A production, followed by its storage in adipose tissue. Elevated levels of retinol (a major form of vitamin A) in the adipose tissue of homozygous rabbits with a deletion in the *BCO2* gene were observed in New Zealand Red rabbits [20,36], but not in crossbred rabbits [16]. They were not noted in the adipose tissue or the brain of Termond White rabbits in the current experiment, either. Apart from retinol, we also measured *TTPA* gene expression and the levels of α-tocopherol (the most biologically active form of vitamin E), because vitamins A and E are non-enzymatic antioxidants [37]. However, quite unexpectedly, higher α-tocopherol levels in the perirenal fat and brain of *BCO2* del/del rabbits were not accompanied by higher retinol concentrations (Table 1 and Table 2). An absence of a simultaneous significant increase in the levels of vitamins A and E in del/del homozygous rabbits was noted in an earlier study [16,20]. The correlation between the concentrations of vitamins A and E in animals remains to be elucidated, but according to Napoli et al. [38], α-tocopherol may play an important role in tissue retinol homeostasis in rats.

A novel finding from this experiment is that in homozygous rabbits with AAT-deletion mutation at codon 248 of the *BCO2* gene (*BCO2* del/del), compared with their counterparts carrying other genotypes (ins/ins and ins/del), the expression of *BCO2*, *LRAT*, and *TTPA* genes may decrease in adipose tissue, and *TTPA* gene expression may decrease in the brain. Moreover, an increase in dietary lutein content in *BCO2* del/del rabbits (to 320 mg/kg) may increase the expression of *BCO1*, *LRAT,* and *TTPA* genes in adipose tissue and the expression of the *BCO2* gene in the brain. Another implication of the study is that the content of lutein, β-carotene, and α-tocopherol increases in both the adipose tissue and brain of *BCO2* del/del rabbits. These findings can promote our understanding of carotenoid metabolism in rabbits as well as other animal species. The gene expression levels noted in the current study indicate that the metabolism of vitamin E, vitamin A, and carotenoids is interrelated, even if it is not reflected in the final levels of these micronutrients in the tissues or organs of animals. Due to the limitations of this study, further research should focus on rabbits of different breeds fed different diets as well as on specialized cell lines. Moreover, gene expression was determined based on only one gene from each metabolic pathway, and the obtained data are incomplete. To address this limitation, other tissues and organs and a larger number of genes involved in metabolic pathways should be analyzed.

## 4. Materials and Methods

All experimental procedures were approved by the Local Ethics Committee in Olsztyn, Poland (consent No. 065.12.2020), and the animals were cared for in accordance with the guidelines of the European Parliament (EC Directive 86/60963/2010 on the protection of animals used for experimental and other scientific purposes).

The rabbits used in this experiment were also involved in the feeding trial described by Strychalski et al. [26]. In brief, sixty male 35-day-old Termond White rabbits were allocated to three groups based on their genotype at codon 248 of the *BCO2* gene (ins/ins, ins/del, and del/del). Each group comprised two subgroups; one subgroup (control group, n = 10) was administered a standard diet, whereas the diet offered to the other subgroup was supplemented with 6 g/kg of Aztec marigold flower extract offered as a dried powder containing 4.96% of lutein and 0.28% of zeaxanthin (marigold group, n = 10). The ingredients and chemical composition of the control diet are shown in Appendix A.

The experiment lasted for 8 weeks, from weaning at 35 days of age until 91 days of age. The initial and final body weights of the rabbits were 726.1 ± 66.09 g and 2411.2 ± 210.99 g, respectively (mean ± SD). The rabbits were kept in individual wire-mesh cages, under standard management conditions. The temperature was maintained within the range of 16–18 °C, and humidity was within the range of 60–75%. The room was intensively ventilated. The photoperiod consisted of 12 h of light and 12 h of darkness.

Genotyping was performed and RNA expression was determined according to the methods described in our previous study [26]; approximately 10 g samples of perirenal fat and the right hemisphere of the brain were analyzed in the current experiment. 

Chemical analyses were described earlier by Strychalski et al. [26]. Briefly, the nutrient content of the feed was determined with AOAC methods [39], and NDF was determined according to the method of Van Soest et al. [40]. The content of retinol and α-tocopherol in feed was analyzed by high-performance liquid chromatography (HPLC), in accordance with Polish Standards (PN–EN ISO 14565 2002; PN–EN ISO 6867 2002). The concentrations of retinol and α-tocopherol in animal tissues were determined as described by Hőgberg et al. [41] and Xu [42], and their levels in feed samples and the liver and blood serum of rabbits were determined by reversed-phase (RP) HPLC.

Calculations were made using R software [43]. The specialized MCMC.qpcr package was used for qPCR analyses [44]. A one-way design model was used to compare three different genotypes within the same diet, and two different diets within the same genotype. Reference genes (*GAPDH* and *β-actin*) were added as priors to the model function to account for variations in cDNA and were allowed 1.2-fold changes [44]. In charts, posterior means and standard errors of the mean (SEM) were plotted as log2-transformed abundance. The data on the content of selected compounds in the adipose tissue and brain of rabbits were processed statistically by analysis of variance (ANOVA) followed by Duncan’s multiple range test. These data are presented in the Tables as means ± standard deviations (SD).

## Figures and Tables

**Figure 1 ijms-24-02304-f001:**
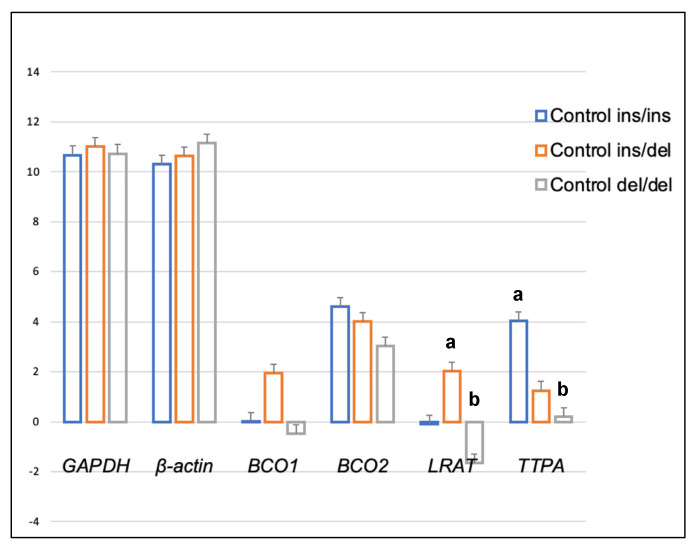
Relative *BCO1*, *BCO2*, *LRAT,* and *TTPA* mRNA levels (log2 abundance) in the perirenal fat of rabbits with different genotypes at codon 248 of the *BCO2* gene (ins/ins vs. ins/del vs. del/del) fed the control diet. *GAPDH* and *ß-actin* were used as reference genes. Data represent posterior means (expressed as arbitrary units) ± SEM (standard error of the mean). Means followed by the different superscripts (a, b) differ significantly (*p* ≤ 0.05).

**Figure 2 ijms-24-02304-f002:**
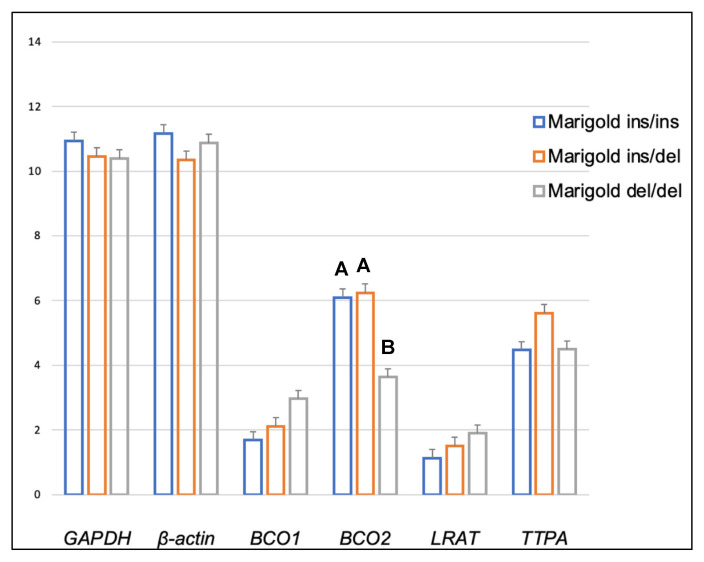
Relative *BCO1*, *BCO2*, *LRAT,* and *TTPA* mRNA levels (log2 abundance) in the perirenal fat of rabbits with different genotypes at codon 248 of the *BCO2* gene (ins/ins vs. ins/del vs. del/del) fed a diet with the addition of Aztec marigold flower extract. *GAPDH* and *ß-actin* were used as reference genes. Data represent posterior means (expressed as arbitrary units) ± SEM (standard error of the mean). Means within a row followed by the different superscripts (A, B) differ highly significantly (*p* ≤ 0.01).

**Figure 3 ijms-24-02304-f003:**
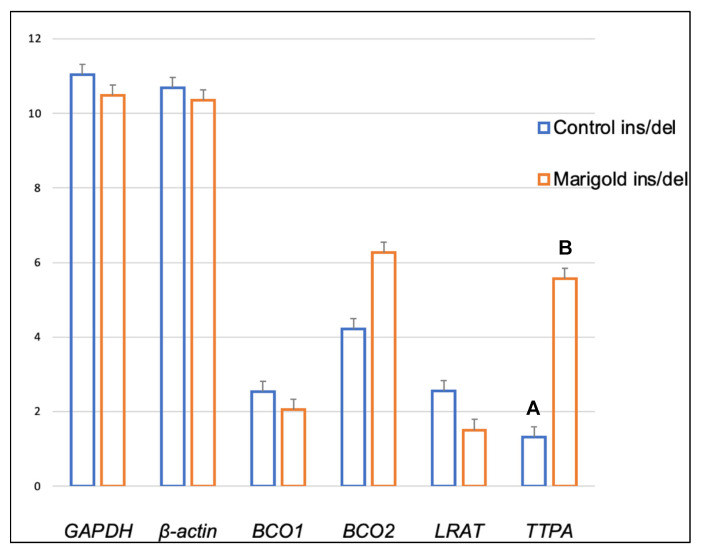
Relative *BCO1*, *BCO2*, *LRAT,* and *TTPA* mRNA levels (log2 abundance) in the perirenal fat of rabbits with the ins/del genotype at codon 248 of the *BCO2* gene, fed different diets (control diet vs. a diet with the addition of Aztec marigold flower extract). *GAPDH* and *ß-actin* were used as reference genes. Data represent posterior means (expressed as arbitrary units) ± SEM (standard error of the mean). Means followed by the different superscripts (A, B) differ highly significantly (*p* ≤ 0.01).

**Figure 4 ijms-24-02304-f004:**
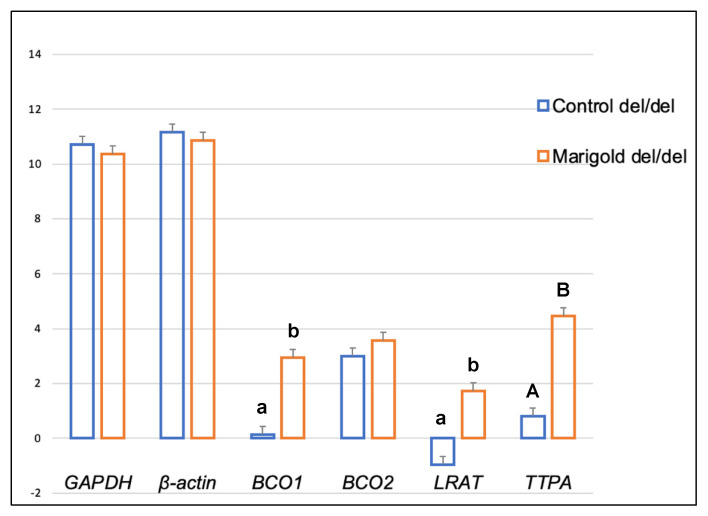
Relative *BCO1*, *BCO2*, *LRAT,* and *TTPA* mRNA levels (log2 abundance) in the perirenal fat of rabbits with the del/del genotype at codon 248 of the *BCO2* gene, fed different diets (control diet vs. a diet with the addition of Aztec marigold flower extract). *GAPDH* and *ß-actin* were used as reference genes. Data represent posterior means (expressed as arbitrary units) ± SEM (standard error of the mean). Means followed by the different superscripts (A, B) differ highly significantly (*p* ≤ 0.01). Means followed by the different superscripts (a, b) differ significantly (*p* ≤ 0.05).

**Figure 5 ijms-24-02304-f005:**
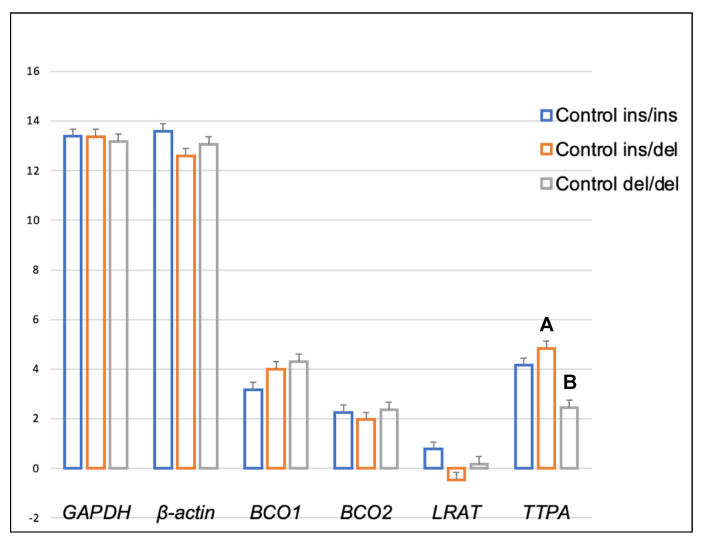
Relative *BCO1*, *BCO2*, *LRAT,* and *TTPA* mRNA levels (log2 abundance) in the brain of rabbits with different genotypes at codon 248 of the *BCO2* gene (ins/ins vs. ins/del vs. del/del) fed the control diet. *GAPDH* and *ß-actin* were used as reference genes. Data represent posterior means (expressed as arbitrary units) ± SEM (standard error of the mean). Means followed by the different superscripts (A, B) differ highly significantly (*p* ≤ 0.01).

**Figure 6 ijms-24-02304-f006:**
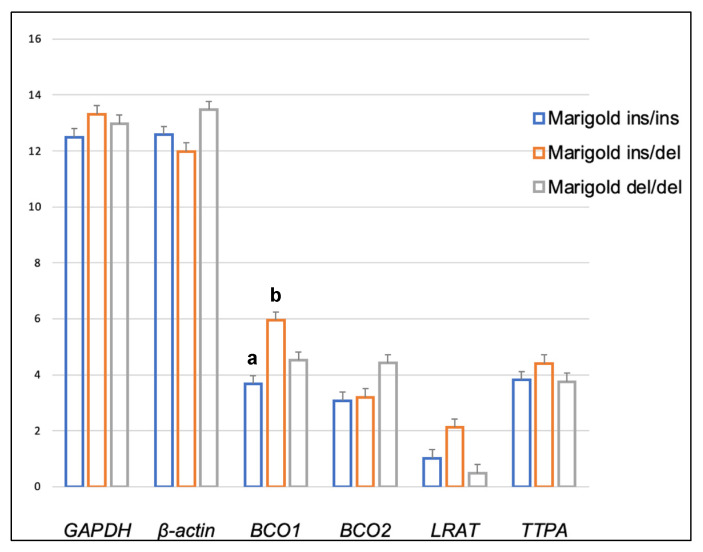
Relative *BCO1*, *BCO2*, *LRAT,* and *TTPA* mRNA levels (log2 abundance) in the brain of rabbits with different genotypes at codon 248 of the *BCO2* gene (ins/ins vs. ins/del vs. del/del) fed a diet with the addition of Aztec marigold flower extract. *GAPDH* and *ß-actin* were used as reference genes. Data represent posterior means (expressed as arbitrary units) ± SEM (standard error of the mean). Means followed by the different superscripts (a, b) differ significantly (*p* ≤ 0.05).

**Figure 7 ijms-24-02304-f007:**
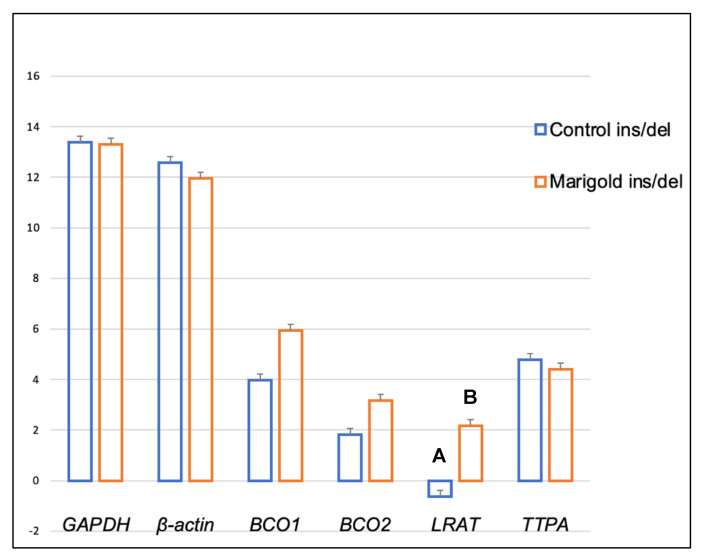
Relative *BCO1*, *BCO2*, *LRAT,* and *TTPA* mRNA levels (log2 abundance) in the brain of rabbits with the ins/del genotype at codon 248 of the *BCO2* gene, fed different diets (control diet vs. a diet with the addition of Aztec marigold flower extract). *GAPDH* and *ß-actin* were used as reference genes. Data represent posterior means (expressed as arbitrary units) ± SEM (standard error of the mean). Means followed by the different superscripts (A, B) differ highly significantly (*p* ≤ 0.01).

**Figure 8 ijms-24-02304-f008:**
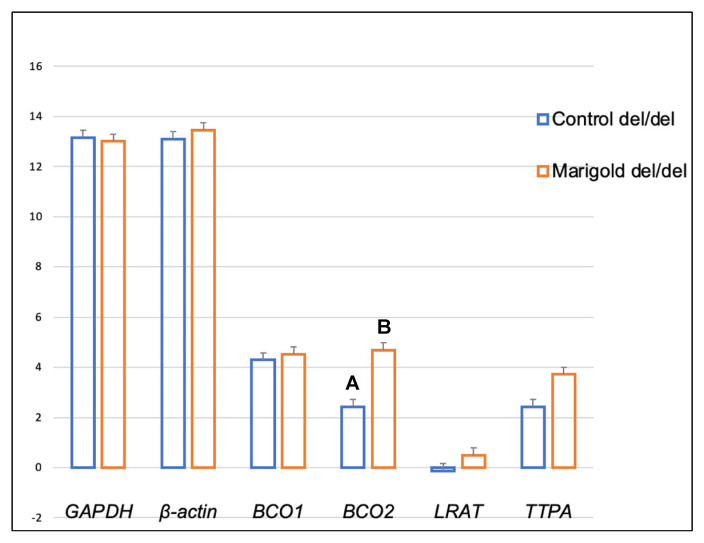
Relative *BCO1*, *BCO2*, *LRAT,* and *TTPA* mRNA levels (log2 abundance) in the brain of rabbits with the del/del genotype at codon 248 of the *BCO2* gene, fed different diets (control diet vs. a diet with the addition of Aztec marigold flower extract). *GAPDH* and *ß-actin* were used as reference genes. Data represent posterior means (expressed as arbitrary units) ± SEM (standard error of the mean). Means followed by the different superscripts (A, B) differ highly significantly (*p* ≤ 0.01).

**Table 1 ijms-24-02304-t001:** Content of selected compounds in the perirenal fat (μg/g) of rabbits (mean ± SD).

Compound	Diet	*BCO2* Genotypes			*p*-Value
		ins/ins	ins/del	del/del	
Lutein	Control	0.06±0.05 ^A^	0.07±0.07 ^A^	0.58±0.24 ^B^	*<0.001*
	Marigold	0.04±0.02 ^A^	0.05±0.05 ^A^	5.37±3.16 ^B^	*<0.001*
*p*-value		*0.304*	*0.655*	*<0.001*	
Zeaxanthin	Control	0.01±0.01 ^A^	0.01±0.01 ^A^	0.05±0.04 ^B^	*<0.001*
	Marigold	0.01±0.01 ^A^	0.01±0.01 ^A^	0.57±0.40 ^B^	*<0.001*
*p*-value		*0.093*	*0.031*	*<0.001*	
β-carotene	Control	0.08±0.04 ^A^	0.13±0.07 ^a^	0.25±0.14 ^Bb^	*<0.001*
	Marigold	0.08±0.04 ^A^	0.08±0.03 ^A^	0.56±0.39 ^B^	*<0.001*
*p*-value		*0.854*	*0.219*	*0.034*	
Retinol	Control	6.48±3.89	6.01±5.41	7.03±6.15	*0.772*
	Marigold	6.19±4.82	6.39±6.55	6.80±4.78	*0.965*
*p*-value		*0.886*	*0.864*	*0.851*	
α-tocopherol	Control	4.02±2.33 ^A^	6.90±3.31	9.63±4.99 ^B^	*0.009*
	Marigold	6.65±3.81 ^a^	7.60±4.15 ^a^	13.32±6.34 ^b^	*0.011*
*p*-value		*0.079*	*0.616*	*0.165*	

Means within a row followed by the different superscripts (A, B) differ highly significantly (*p* ≤ 0.01). Means within a row followed by the different superscripts (a, b) differ significantly (*p* ≤ 0.05).

**Table 2 ijms-24-02304-t002:** Content of selected compounds in the brain (μg/g) of rabbits (mean ± SD).

Compound	Diet	*BCO2* Genotypes			*p*-Value
		ins/ins	ins/del	del/del	
Lutein	Control	0.15±0.05^A^	0.15±0.07^A^	0.66±0.47^B^	*<0.001*
	Marigold	0.18±0.18^A^	0.26±0.16^A^	1.89±1.39^B^	*<0.001*
*p*-value		*0.662*	*0.054*	*0.016*	
Zeaxanthin	Control	0.02±0.01^a^	0.01±0.01^A^	0.04±0.02^Bb^	*<0.001*
	Marigold	0.01±0.01^A^	0.01±0.01^A^	0.21±0.20^B^	*<0.001*
*p*-value		0.153	0.542	<0.001	
β-carotene	Control	0.06±0.04^a^	0.09±0.05	0.13±0.06^b^	*0.019*
	Marigold	0.07±0.03^A^	0.07±0.04^A^	0.21±0.11^B^	*<0.001*
*p*-value		0.520	0.320	0.074	
Retinol	Control	0.33±0.24	0.28±0.22	0.39±0.37	*0.691*
	Marigold	0.53±0.28	0.68±0.46	0.73±0.46	*0.455*
*p*-value		0.100	0.025	0.052	
α-tocopherol	Control	0.00±0.00^A^	0.00±0.00^A^	0.04±0.02^B^	*<0.001*
	Marigold	0.00±0.00^A^	0.00±0.00^A^	0.02±0.01^B^	*<0.001*
*p*-value		0.938	0.964	0.002	

Means within a row followed by the different superscripts (A, B) differ highly significantly (*p* ≤ 0.01). Means within a row followed by the different superscripts (a, b) differ significantly (*p* ≤ 0.05).

## Data Availability

The authors declare that the data of this research are not deposited in an official repository and data will be available upon reasonable request.

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
