# Peer review of "The BCO2 Genotype and the Expression of BCO1, BCO2, LRAT, and TTPA Genes in the Adipose Tissue and Brain of Rabbits Fed a Diet with Marigold Flower Extract"

_ijms, 2023, doi:10.3390/ijms24032304_

Round 1

Reviewer 1 Report

General-Major comments

Introduction: what is missing is why to address the specific gap or why there is a need to investigate the effect of genotype and marigold on the gene expression in rabbits specifically. Meanwhile, there is not any reference to other dietary practices-ingredients followed thus far to enrich the diet of rabbits in carotenoids. For example, rabbit diets may contain as basal ingredient various quantities of alfalfa which can also increase the carotenoids content in the diet.

L244-246: or could it be because the subjects-rabbits in the current experiment were not exposed to pro-oxidative procedures?

Minor comments

L235-236: could the authors elaborate more on this conclusion-hypothesis? Could there be a distinction between target-peripheral tissues and liver? Meanwhile, the role of adipose tissue in metabolism is not negligible and especially regarding fat metabolism is rather important.

Author Response

Dear Reviewer,

Thank you very much for reading our manuscript and providing your comments. We read them carefully, and the manuscript was revised accordingly. All changes were marked in the revised manuscript using the “Track Changes” mode. Please find our answers below.

Reviewer’s comment:

General-Major comments

Introduction: what is missing is why to address the specific gap or why there is a need to investigate the effect of genotype and marigold on the gene expression in rabbits specifically.

Response:

- Thank you for this suggestions. We have tried to complete the information regarding the purpose of the work (lines 86-87). This also goes along with another suggestion you made regarding the discussion (lines 368-376).

Reviewer’s comment:

Meanwhile, there is not any reference to other dietary practices-ingredients followed thus far to enrich the diet of rabbits in carotenoids. For example, rabbit diets may contain as basal ingredient various quantities of alfalfa which can also increase the carotenoids content in the diet.

Response:

- We added information regarding the addition of other sources of carotenoids to the diet of rabbits (lines 76-79). As it is known, fresh green fodder (from alfalfa or even grass) is a very good source of carotenoids, while the process of drying them to obtain dried material causes the degradation of a significant amount of carotenoids. This is the reason why marigold flower extract can be a very attractive addition to granules for rabbits, other animals or for humans.

Reviewer’s comment:

L244-246: or could it be because the subjects-rabbits in the current experiment were not exposed to pro-oxidative procedures?

Response:

- We added this idea (lines 386-388). Thank you!

Reviewer’s comment:

L235-236: could the authors elaborate more on this conclusion-hypothesis? Could there be a distinction between target-peripheral tissues and liver? Meanwhile, the role of adipose tissue in metabolism is not negligible and especially regarding fat metabolism is rather important.

Response:

We agree with the Reviewer, so we extended our thought (lines 368-376). We also believe that the role of adipose tissue in the body cannot be overestimated.

We would also like to mention that at the request of another reviewer, we changed the type of graphs to bar graphs and used SEM as a measure of volatility. In addition, any differences between the averages were previously marked with the same letter, while in the current version they are two different letters.

Once again, we would like to thank the Reviewer for all valuable comments which have led to significant improvements in the manuscript. We hope we responded to each comment with due care.

Reviewer 2 Report

The manuscript presents an interesting topic and falls on the IJMS. However, I recommend working on some of the sections before the manuscript is accepted for publication:

. M&M should describe in more detail the chemical and genetic analyses performed, and explain the handling of the animals (how long the experiment lasted, what was the weight of the animals when entering and leaving the experiment, did all animals have the same amount of fat, etc.) and the environmental housing conditions (temperature and humidity were constant, etc.).

. Results. The figures are a bit confusing for me because of the letters. I would like to point out to the authors that different letters are always used to indicate differences within and between groups (no similar letters). Looking at their figures it is not clear to me which groups differ from each other because credible intervals are solaping. Could authors change the credible intervals for the standard errors?. Please, check the figures and the abcrition of the letters.

Author Response

Dear Reviewer,

Thank you very much for reading our manuscript and for your suggestions. We reviewed them and revised the manuscript accordingly. All changes were made in "Track changes" mode. Below please find our answers to your suggestions.

Reviewer’s comment:

M&M should describe in more detail the chemical and genetic analyses performed, and explain the handling of the animals (how long the experiment lasted, what was the weight of the animals when entering and leaving the experiment, did all animals have the same amount of fat, etc.) and the environmental housing conditions (temperature and humidity were constant, etc.).

Response:

- We thank the Reviewer for this comment. We expanded the information in this chapter a bit (we added lines 453-458 and 462-483). The full version of M&M is in our previous publication, which is open to everyone (Open Access), so in order not to commit self-plagiarism, the information in this chapter is, of course, abbreviated. However, if you feel that certain aspects should be still extended, we will do so.

Reviewer’s comment:

Results. The figures are a bit confusing for me because of the letters. I would like to point out to the authors that different letters are always used to indicate differences within and between groups (no similar letters). Looking at their figures it is not clear to me which groups differ from each other because credible intervals are solaping. Could authors change the credible intervals for the standard errors?. Please, check the figures and the abcrition of the letters.

Response:

- We agree with the Reviewer, so we changed the charts to bar charts with SEM variability for better readability. In addition, we changed the differences between the groups from the same letters to different letters, both in charts and tables.

Once again, we would like to thank the reviewer very much for reviewing our manuscript. We hope that the improvements we have made are satisfactory.

authors

Round 2

Reviewer 1 Report

The authors have modified and changed the manuscript according to comments/suggestions.